# M^pro^pred: A machine learning (ML) driven Web-App for bioactivity prediction of SARS-CoV-2 main protease (M^pro^) antagonists

Nadim Ferdous[1⊙], Mahjerin Nasrin Reza[1⊙], Mohammad Uzzal Hossain[2,3], Shahin Mahmud[1], Suhami Napis[4], Kamal Chowdhury[5], A. K. M. Mohiuddin[1]*

1 Department of Biotechnology and Genetic Engineering, Mawlana Bhashani Science and Technology University, Santosh, Tangail, Bangladesh, 2 Department of Pharmacology, Medical Sciences Division, University of Oxford, Oxford, United Kingdom, 3 Bioinformatics Division, National Institute of Biotechnology, Ashulia, Ganakbari, Savar, Dhaka, Bangladesh, 4 Department of Molecular Biology, Universiti Putra Malaysia, Serdang, Selangor D.E., Malaysia, 5 Biology Department, Claflin University, Orangeburg, SC, United States of America

⊙ These authors contributed equally to this work.
* akmmohiu@yahoo.com

**Data Availability Statement:** All relevant data are within the paper and its Supporting Information files.

## Abstract

The severe acute respiratory syndrome coronavirus 2 (SARS-CoV-2) pandemic emerged in 2019 and still requiring treatments with fast clinical translatability. Frequent occurrence of mutations in spike glycoprotein of SARS-CoV-2 led the consideration of an alternative therapeutic target to combat the ongoing pandemic. The main protease (M^pro^) is such an attractive drug target due to its importance in maturating several polyproteins during the replication process. In the present study, we used a classification structure–activity relationship (CSAR) model to find substructures that leads to to anti-M^pro^ activities among 758 nonredundant compounds. A set of 12 fingerprints were used to describe M^pro^ inhibitors, and the random forest approach was used to build prediction models from 100 distinct data splits. The data set's modelability (MODI index) was found to be robust, with a value of 0.79 above the 0.65 threshold. The accuracy (89%), sensitivity (89%), specificity (73%), and Matthews correlation coefficient (79%) used to calculate the prediction performance, was also found to be statistically robust. An extensive analysis of the top significant descriptors unveiled the significance of methyl side chains, aromatic ring and halogen groups for M^pro^ inhibition. Finally, the predictive model is made publicly accessible as a web-app named M^pro^pred in order to allow users to predict the bioactivity of compounds against SARS-CoV-2 M^pro^. Later, CMNPD, a marine compound database was screened by our app to predict bioactivity of all the compounds and results revealed significant correlation with their binding affinity to M^pro^. Molecular dynamics (MD) simulation and molecular mechanics/Poisson Boltzmann surface area (MM/PBSA) analysis showed improved properties of the complexes. Thus, the knowledge and web-app shown herein can be used to develop more effective and specific inhibitors against the SARS-CoV-2 M^pro^. The web-app can be accessed from https://share.streamlit.io/nadimfrds/mpropred/Mpropred_app.py.

**Funding:** The author(s) received no specific funding for this work.

**Competing interests:** The authors have declared that no competing interests exist.

## Introduction

The COVID-19 pandemic, which was triggered by SARS-CoV-2, is still having a disastrous impact on public health and the worldwide economy [1,2]. On earlier march of 2020, the outbreak was declared as a pandemic after initially discovering the virus in end of 2019 in Wuhan, China [3,4]. SARS-CoV-2 is a single-stranded RNA virus with an increased mutation rate, a short period of replication, and a high production of virion [5–8]. The virus acquires a significant amount of genetic variation as it spreads, enabling it to adapt quickly to stresses brought on by natural selection, particularly those imposed by the immune system of the host. Mutations build up over time, resulting in alterations to the amino acids that make immune-targeted proteins less antigenic. This process is known as "antigenic drift", that is the gradual alteration in viral protein antigenicity caused by selective immunological pressure [9]. Antigenic drift permits viruses to avoid host immunity continuously, allowing for recurring viral outbreaks. In cases of acute infectious disease, antibody responses are mostly responsible, resulting in the selection of escape mutants [9]. The spike protein contains several variations in amino acids identified in SARS-CoV-2 variants, is the main target that antibodies neutralize [10]. These antibodies are the sole immune system component capable of providing sterilizing immunity, preventing infection of host cells by the virus. The SARS-CoV-2 spike protein has evolved at a considerably faster rate than similar proteins in additional known viruses that cause severe infectious diseases in humans [11]. In addition, SARS-CoV-2 proteins have accumulated a substantial number of amino acid modifications that are not recognized antibody targets. In acute viral infections, antibody responses are predominantly responsible for antigenic drift; therefore, these amino acid alterations may have given the virus a fitness benefit independent to antibody immunity [12,13].

As a result, structural and functional research of SARS-CoV-2 infection processes have primarily shifted to the main protease (M$^{pro}$), which cleaves native polypeptides and forms active fragments that are crucial for viral replication, transcription, and translation process [14]. The protease is consists of three domains [15]. A loop (residues 185–200) connects domain II with domain III. The ligand binding site is found in the loop between the first two domains, where the catalytic dyad consisting of Cys 145 and His 41 is crucial for ligand management [15]. SARS-CoV-2 replication is significantly facilitated by M$^{pro}$ [15]. Moreover, it is not functionally associated with human homologue proteases, implying that M$^{pro}$ is a promising target for therapeutic development [15].

The functions of compounds both at structural and chemical level, are crucial in understanding the impact of physicochemical qualities on bioactivity. Computer-aided drug design (CADD) portrays a set of computational techniques that has proven useful in chemical biology and computational approaches to understanding the structure–activity relationship [16,17]. Computational tools are used to decipher bioactivity using ligands, which are known as chemical descriptors [18]. Molecular descriptors computation softwares can also be used to calculate the physicochemical properties of different compounds. The QSAR (quantitative structure activity relationship) is considered a widely used computational technique to construct prediction models which can distinguish the impact of important molecular fingerprints regulating their bioactivities and properties [19]. A variety of targets, including antioxidant [20], antibacterial [21,22], anticancer [23,24], and antiviral [25,26] activities, have been successfully modeled using QSAR models.

In silico strategies such as QSAR, pharmacophore modeling, docking and molecular dynamics (MD) simulation have extensively studied for identifying new inhibitors of M$^{pro}$. Isabela et al. used an in-house designed machine learning technique, molecular docking, MM-PBSA calculations, and meta dynamics to find FDA-approved compounds that could

potentially suppress the enzyme activity of the $M^{pro}$ [27]. Nedra et al. developed a machine learning approach by employing the support vector machine (SVM) classification model to categorize two hundred novel chemo-types as potentially active against the viral protease using a dataset of two million commercially accessible drugs [28]. Mahesha et al. used an integrative strategy to screen 1528 anti-HIV1 compounds, using a machine learning predictive model, molecular docking, and a deep learning model that considered the IC50 values of known inhibitors [29]. But the models developed in these works are not accessible as readily available web-apps for the scientific community to further apply on different sets of compounds in order to identify more potent anti-$M^{pro}$ inhibitors.

In the present study, we collected a set of compounds (non-redundant) with known IC50 values against the SARS-CoV-2 $M^{pro}$. Several fingerprint descriptors were used to describe the examined compounds binning the IC50 values to qualitative labels such as active and inactive. Following that, the RF algorithm was used to build prediction models. The sensitivity, specificity, accuracy and Matthews correlation coefficient of the built QSAR model were tested in classifying active or inactive compounds against SARS-CoV-2 $M^{pro}$. In addition, the underlying key substructures that are critical for bioactivity were identified and defined. A web-app was built based on the model and made publicly accessible. We also utilized our app to view into the correlation of predicted bioactivity of compounds with their binding affinity to $M^{pro}$. As a result, this knowledge can be exploited to develop more potent and specialized drugs against SARS-CoV-2 $M^{pro}$.

## Materials and methods

A step-wise protocol was followed to build a web-app in order to predict bioactivities of compounds against the $M^{pro}$ of SARS-CoV-2. The work flow is shown in Fig 1.

### Dataset preparation and curation

A data set consisting of antagonists against SARS-CoV-2 main protease ($M^{pro}$) was compiled from an extensive literature review that was initially comprised of 758 compounds [30]. The mean value was calculated in the event that multiple IC50 values were found for the same compound. As our study aims to developing a classification model of $M^{pro}$ antagonists, we defined the thresholds as <0.5 and >10 μM to distinguish active compounds from inactive ones, respectively. Also, intermediate bioactivities with IC50 values that ranged between 1 and 10 were excluded from the study, consisting of 284 inhibitors. Finally, the curated set of compounds consisting of 478 inhibitors was obtained and analyzed.

### Calculation of molecular descriptors

The PaDEL-Descriptor software was utilized briefly to compute the fingerprints of the data set [31]. Generally, molecular descriptors are very crucial for QSAR studies because they are used to characterize the various properties of compounds and aid in the structural information analyses. In the present study, 12 molecular fingerprints that belong to 9 classes were used to describe the structures, and these consisted of 2D AtomPairs, CDK (including extended and graph only version), E-state, PubChem, Klekota–Roth, Substructure and MACCS.

### Data filtering and balancing

To choose the fingerprint descriptor sets, variables that were constant or nearly constant were used with a view to removing the bias and complexity in building the model. Using 0.1 as the SD cut-off value, all the constants of each fingerprint descriptor were calculated. For further

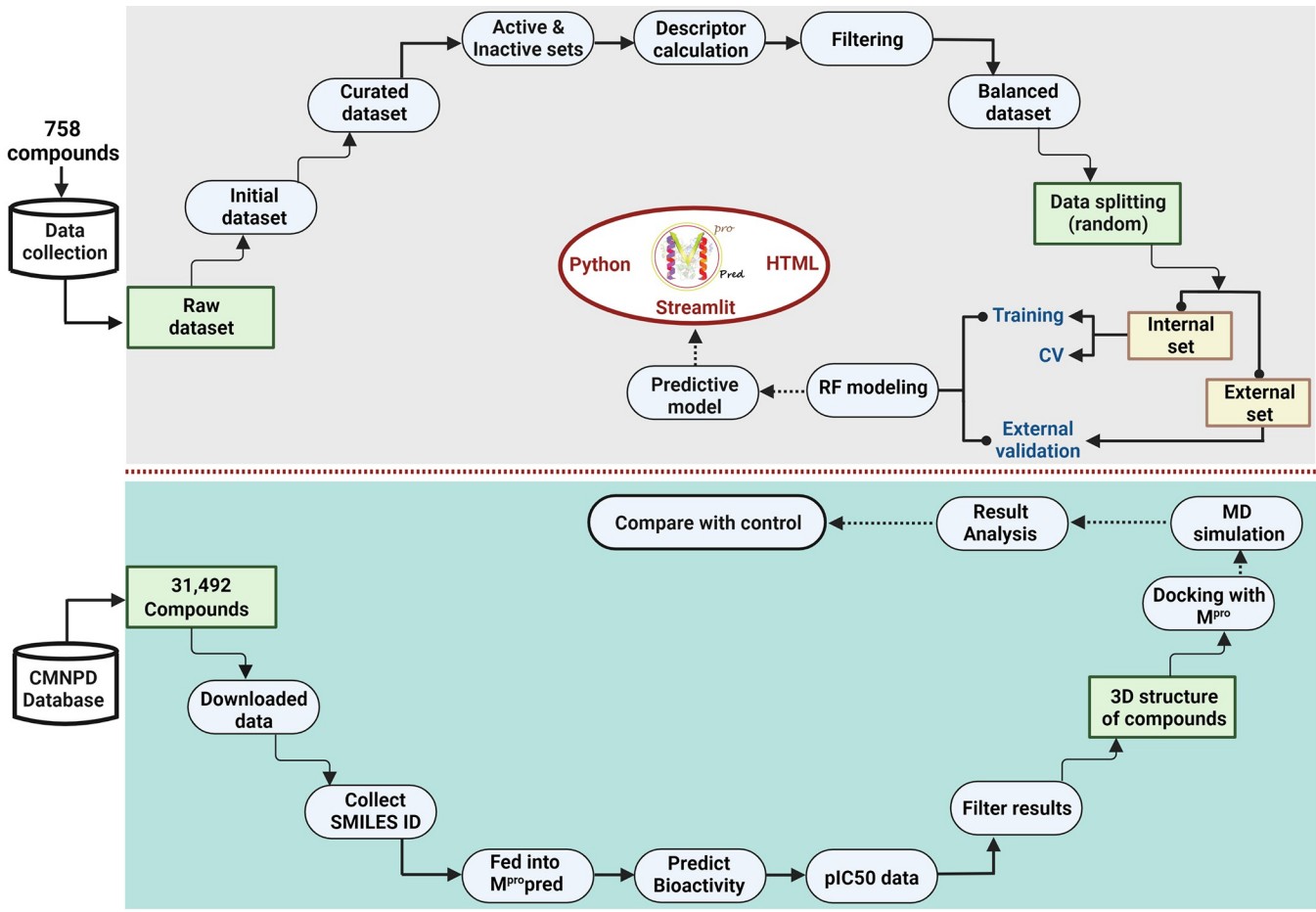

**Fig 1. Schematic workflow of building a web-app to predict bioactivities of compounds against the M$^{pro}$ of SARS-CoV-2.** The upper side of the flowchart depicts the methodology of building the web-app while the lower side shows the molecular modeling protocol that was used to view into the correlation of predicted bioactivity of compounds with their corresponding binding affinity.

investigation, fingerprints having SD values of >0.1 were chosen. The undersampling technique was employed by random selection of subset of the active compounds from the starting set to avoid the propensity for overfitting of imbalanced data. Additionally, the data was divided into two sets, with the internal set being 80% of the total data set and the external set comprising the remaining 20% to avoid any possibility of getting a predicted model that is biased.

## Multivariate analysis of model

A CSAR model's prediction performance is influenced by both the predictor and the compound descriptors. Considering the success in various models and the interpretability in many applications, we employed RF in this study.

Random forest is an ensemble classifier that uses a randomly selected subset of training samples and variables to generate a number of decision trees. The classification of RF begins at the root node, where the value of particular descriptors is used to divide the data set at every node, with the descriptors of various activities being primarily transported to distinct branches [32,33]. The classification is then obtained by averaging the outcomes of all trees using a majority vote from each tree [34,35]. The *randomForest* package of the R language was used to

create the RF classifier. To effectively predict $M^{pro}$ inhibitor activity, two RF model parameters must be tuned: the number of trees used to form the RF classifier (ntree) and the number of random candidate features (*mtry*). The parameter "*mtry*" was created using the *randomForest* package's tuneRF function, while the ntree parameter was tuned using a 10-fold CV technique from the range of ntree $\in$ {100,200,...,1,000} [36]. The importance estimator, an efficient built-in component of the RF model, was also utilized to find informative descriptors to better explain the bioactivity of $M^{pro}$ inhibitors.

## Modelability of data set

The underlying relatedness of chemical structures and their bioactivities is required for modelability. Activity cliffs, also known as two compounds with remarkably different bioactivities (i.e., one pair of compounds has favorable biological activity while the other in the pair has low bioactivity), are detrimental to machine learning algorithms that try to correlate structures with related biological activity. Similar compounds having comparable bioactivities would, on the other hand, contribute favorably to the data set's modelability. Golbraikh et al. developed this modelability index (MODI) [37]. The following formula can be used to calculate the statistical metric:

Step 1: The normalized Euclidean distance (Dnormalized) for each pair of the compounds, Ci and Cj described by m-dimensional vector is calculated as follows:

$$d_{ij} = ||C_i - C_j|| = \sqrt{\sum_{k=1}^{m} (C_{ik} - C_{jk})^2} \tag{1}$$

$$\bar{d}_i = \frac{\sum_{j=1}^{n} d_{ij}}{n-1} \tag{2}$$

$$\bar{D}_{normalized} = \frac{\bar{D} - \min(\bar{D})}{\max(\bar{D}) - \min(\bar{D})} \tag{3}$$

where $d_{ij}$, $\bar{d}_i$ shows distance scores between two compounds and the n represent mean Euclidean distance.

Step 2: The MODI can be calculated for each compound in a data set by determining whether its first nearest neighbor belongs to the same class as the compound or a different class:

$$MODI = \frac{1}{N_c} \sum_{I=1}^{N_c} \frac{N_i^{same}}{N_i^{total}} \tag{4}$$

Where the $N_C$ denotes the number of classes, $N_i^{same}$ denotes the number of total compounds in the *i*th class having the same *i*th class as their first nearest neighbors, and $Ni^{total}$ denotes number of total compounds in the *i*th class. Any data set is deemed modelable provided that MODI index falls beyond the cutoff value of 0.65. Here, the MODI index was calculated using a R code that was used for assessing modelibility of the HCVpred [38] server.

## Model validation (Statistical approach)

Several statistical measures, such as overall prediction sensitivity (Sn), specificity (Sp), accuracy (Ac) and Matthew's correlation coefficient (MCC), were used to evaluate the model's fitness.

$$\text{Sn} = \frac{TP}{(TP + FN)} \times 100 \tag{5}$$

$$Sp = \frac{TN}{(TN + FP)} \times 100 \tag{6}$$

$$Ac = \frac{TP + TN}{(TP + TN + FP + FN)} \times 100 \tag{7}$$

$$MCC = \frac{TP \times TN - FP \times FN}{\sqrt{(TP + FP)(TP + FN)(TN + FP)(TN + FN)}} \tag{8}$$

where True positives, false positives, true negatives, and false negatives are denoted by the abbreviations TP, FP, TN and FN, respectively.

## Applicability domain analysis

The boundaries within which the model may produce precise predictions for compounds based on similarity towards the compounds on which the model was built are established by the applicability domain (AD). Only those compounds are found inside the AD that match the model's parameters. In this study, the AD of the compounds from both the training and testing sets were analyzed using the PCA bounding box.

## Deployment of model as web-app

Finally, we deployed the developed RF model as a web-app with a view to enabling easy access for the research community. The web-app named "M$^{pro}$pred" was built in the Streamlit python package (https://www.streamlit.io/) and deployed on the "Streamlit Share" cloud application platform while the source-code is maintained in a GitHub repository. The web-app can accept SMILES IDs and compound names in the form of a text (.txt) file and return the predicted pIC50 values of the compounds.

## Correlation of predicted bioactivity with binding affinity (Molecular modeling and simulation)

We further tested the correlation of predicted bioactivity of compounds with their corresponding binding affinity to M$^{pro}$ via an integrated molecular modeling and simulation approach with the utilization of our developed web-app. A new comprehensive marine natural products database named CMNPD was used for this purpose [39]. As no previous research was published on testing the efficacy of the compounds from this database against M$^{pro}$, we downloaded all the available 31,492 compounds from the database, collected their SMILES IDs, and submitted them to the M$^{pro}$pred web-app for bioactivity (pIC50) prediction. Later, the 3D structures of the top compounds with high pIC50 values were generated using Open Babel and prepared for molecular docking upon energy minimization using the MMFF94 forcefield. The 3D-structure of SARS-CoV-2 M$^{pro}$ in complexed with an inhibitor N3 (PDB ID: 6LU7) was used as receptor. Molecular docking was run using Autodock Vina with the same grid box parameters covering the ligand binding residues that were used in our previous work [40,41]. The exhaustiveness value was set to 100. The aim of this approach was to assess whether the compounds with predicted high pIC50 bind to the protease with high affinity.

The top 5 complexes with high binding interaction with M$^{pro}$ were subjected to MD simulation to view their conformational changes. The GROningen MAchine for Chemical Simulations (GROMACS) version 5.1.2 was utilized to perform the MD simulations with the

parameters that we previously used [42,43]. The topologies of proteins and the ligands were generated using the 'pdb2gmx' script and the PRODRG server, respectively [44]. The GRO-MOS96 54a7 force field was used to get the energy minimized conformations of complexes and, further, they were solvated in a square box with 1.0 nm of padding using a single point charge (SPC) water model [45]. The net charges in the systems were neutralized using the 'gmx genion' script of GROMACS. The steepest descent algorithm was employed to minimize energy of the complexes with $< 10.0$ kJ/mol force and a maximum of 50,000 steps. Later, NVT and NPT ensembles were performed to equilibrate the systems, both at 300 K temperature and 1 atm for 100 picoseconds (ps). In the simulation, the thermostat and barostat were chosen as the V-rescale and Parrinello-Rahman, respectively. The final production run was performed for 100 nanoseconds (ns) in the HPC cluster of National Institute of Biotechnology, Savar, Bangladesh at 300 K with a 2-fs time step. The simulations were accelerated using a "NVIDIA GTX 3070" graphics processing unit (GPU). The root mean square deviation (RMSD), root mean square fluctuation (RMSF), radius of gyration (Rg), solvent accessible surface area (SASA) and number of hydrogen bonds were analyzed to evaluate the stability of the complexes after completion of the simulation. The GRACE software was used to plot the graphs.

We also calculated the binding free energies (MM/PBSA) using the 'g_mmpbsa' package of GROMACS followed after the final production run [46]. The following equation is used to calculate the binding energies in this method:

$$\Delta G_{binding} = G_{complex} - (G_{protein} + G_{ligand}) \tag{9}$$

where $\Delta G_{binding}$ is the overall binding energy of the complex, $G_{protein}$ is the free protein binding energy, and $G_{ligand}$ is the unbounded ligand binding energy.

## Results

### Chemical space analysis

The dataset that we used in this study is contained in S1 File including the SMILES ID of the 758 compounds with references. Exploration of the typical distinctions between active and inactive compounds is one of the major motives for undertaking chemical space analysis. We visualized the actives and inactives distribution as the function of molecular weight (MW) vs. the Ghose–Crippen–Viswanadhan octanol–water partition coefficient to investigate the general chemical space (ALogP). Then, using Lipinski's rule-of-five (Ro5), we compared the actives and inactives. Fig 2(A) depicts the MW as the function of ALogP. As can be seen, the majority of the compounds are located in the MW range of 250–600 Da and have an ALogP of 0–6. (Fig 2B–2E) includes visualization of data and the statistical analysis of the Ro5. The majority of the compounds meet the Ro5 criteria, having a MW of 500 Da, nHBDon and nHBAcc and ALogP values of $<10$. Furthermore, the findings of statistical analysis show a noteworthy difference among the active and inactive compounds from employment of the Mann–Whitney U test (Table 1). The ALogP values of inactive compounds were found to be higher than the active ones. The nHBDon values of both the active and inactive compounds were similar, however the nHBAcc values of the active compounds were found to be lower than the values of inactive ones.

Furthermore, the AD of the built model was determined using the MACCS fingerprints as the starting input for PCA analysis, as shown in Fig 2(F). After balancing, the data set of 478 compounds was randomly divided into internal and external (80% and 20% respectively) subsets. It's important to note that the internal set (80%) is used as the training set for building predictive models to predict on the external set. The external set's chemical space distribution was revealed to be well inside the internal set's boundaries. As a result, the AD for the CSAR model described herein appears to be well defined.

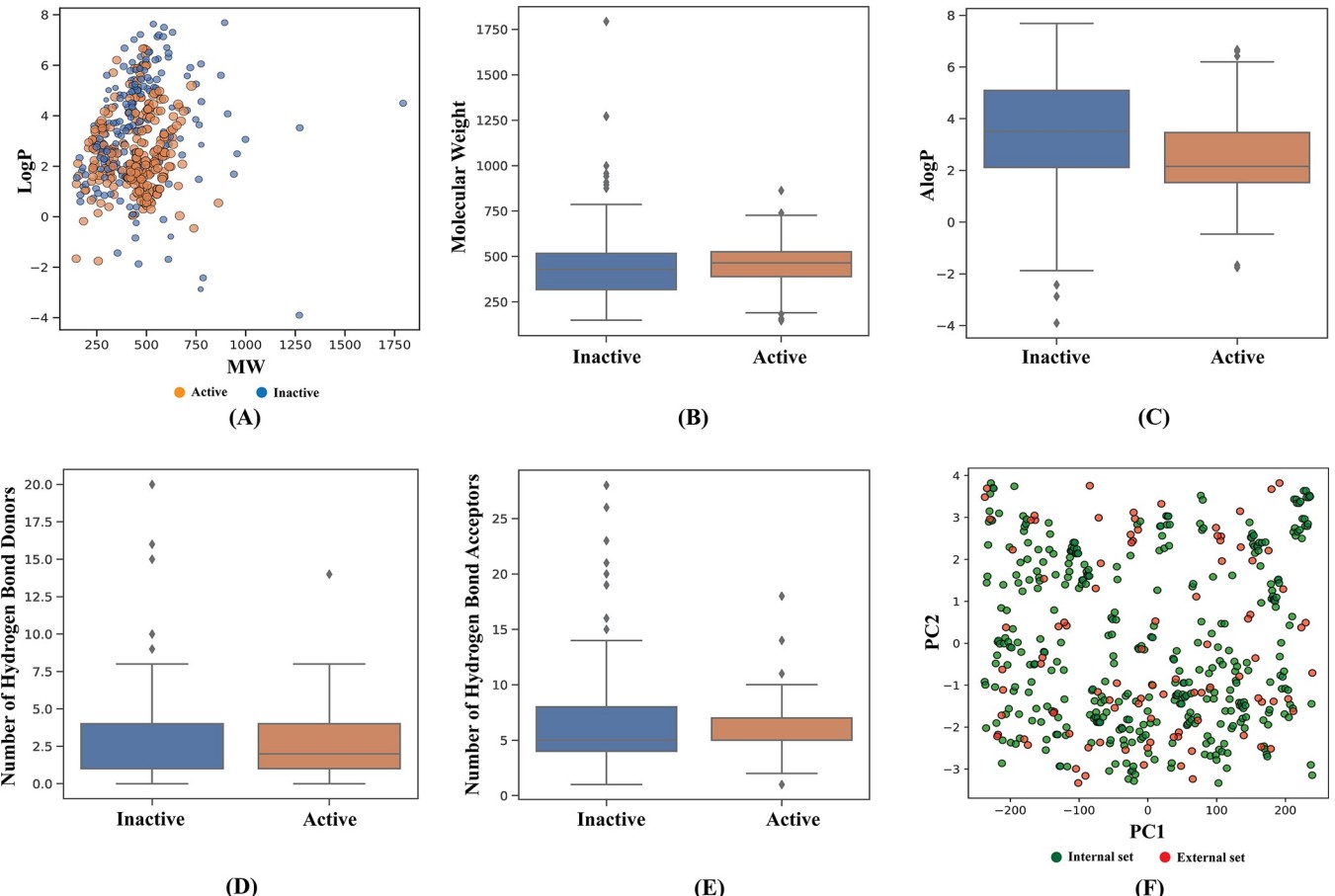

**Fig 2.** Chemical space analysis (A), box plot of Lipinski's rule-of five descriptors (B-E) and applicability domain analysis (F) for analyzed M$^{pro}$ inhibitors.

## QSAR modeling

To develop an interpretable QSAR model, we used fingerprints computed through the PaDEL-Descriptor software. S2–S13 Files contains the computed 12 molecular fingerprints of our dataset. The data set's modelability score or MODI index sorts active compounds from inactive compounds to determine the likelihood of obtaining the CSAR model. It was found that all the fingerprint descriptors have a MODI value greater than 0.65, proving that the data set is reliable for developing a classification model. Table 2 lists all of these fingerprints, as well as their descriptions and MODI indices.

To distinguish between active and inactive M$^{pro}$ inhibitors, we created the CSAR model using the RF algorithm in this work. Table 3 displays the results of 100 independent runs with

**Table 1. Mann–Whitney U test results of various properties of the compounds.**

| Properties | Mann–Whitney U test result | P value |
|---|---|---|
| Molecular weight (MW) | Different distribution | 0.016141739 |
| Octanol–water partition coefficient (AlogP) | Different distribution | 4.12E-10 |
| Number of Hydrogen bond donors (nHBDon) | Different distribution | 0.033026915 |
| Number of Hydrogen bond acceptors (nHBAcc) | Same distribution | 0.095506873 |

**Table 2. List of molecular fingerprints employed in the current study for representing chemical structures of the M^pro^ inhibitor dataset along with their MODI indices.**

| Fingerprint | Number of features | Description | MODI Index | References |
|---|---|---|---|---|
| CDK | 1,024 | Fingerprint having length of 1,024, with a search depth of 8 | 0.77 | [47] |
| CDK extended | 1,024 | Extends the CDK fingerprint with additional bits that describes ring features | 0.78 | [47] |
| CDK graph only | 1,024 | A special CDK version considering only the connectivity, not bond order | 0.77 | [47] |
| E-state | 79 | Atom types of electrotopological state | 0.73 | [48] |
| MACCS | 166 | MACCS keys defined binary representation of chemical features | 0.79 | [49] |
| PubChem | 881 | PubChem defined binary representation of substructures | 0.75 | [50] |
| Substructure | 307 | Presence of the SMARTS patterns for various functional groups | 0.76 | [51] |
| Substructure count | 307 | Count of the SMARTS patterns for various functional groups | 0.79 | [51] |
| Klekota–Roth | 4,860 | Presence of various chemical substructures | 0.79 | [52] |
| Klekota–Roth count | 4,860 | Count of various chemical substructures | 0.75 | [52] |
| 2D atom pairs | 780 | Presence of atom pairs at various topological distances | 0.79 | [53] |
| 2D atom pairs count | 780 | Count of atom pairs at different topological distances | 0.73 | [53] |

all the distinct categories of fingerprints, including internal validation test, 10-fold CV, and external validation test. Best averaged values for the MACCS fingerprints were Ac 84.69% and MCC 0.691, as determined by a 10-fold CV analysis. The external validation for the MACCS, Klekota–Roth, and 2D atom pairs fingerprint descriptors, as shown in Table 3, was also better than the rest of the descriptors. Taking into account the results from 10-fold CV as well as the external validation tests, it is found that the MACCS fingerprint descriptors outperform the other fingerprint classes. Fig 3 contains the plot of experimental vs predicted pIC50 values for model that was constructed with MACCS fingerprint descriptors.

## Interpretation of feature importance

The top-ranked MACCS fingerprints as obtained from the RF model are mentioned in Table 4, comprised of fingerprints pertaining to different classes such as aromatic compounds, nitrogen-containing compounds, oxygen-containing compounds, halogens etc.

**Table 3. Performance summary of CSAR models for predicting M^pro^ inhibitors of SARS-CoV-2.**

| Descriptor class | Training set | | | | | 10-fold CV set | | | | External set | | | |
|---|---|---|---|---|---|---|---|---|---|---|---|---|---|
| | Ac (%) | RMSE | Sn | Sc | MCC | Ac (%) | Sn | Sc | MCC | Ac (%) | Sn | Sc | MCC |
| CDK | 99.79 | 0.1045 | 0.998 | 0.99 | 0.996 | 83.43 | 0.834 | 0.864 | 0.670 | 76.84 | 0.768 | 0.744 | 0.537 |
| CDK extended | 99.58 | 0.107 | 0.996 | 1 | 0.991 | 81.55 | 0.816 | 0.834 | 0.630 | 83.15 | 0.832 | 0.826 | 0.663 |
| CDK graph only | 99.16 | 0.1124 | 0.992 | 0.99 | 0.983 | 84.06 | 0.841 | 0.884 | 0.686 | 84.21 | 0.842 | 0.840 | 0.684 |
| E-state | 96.43 | 0.1247 | 0.964 | 0.96 | 0.927 | 88.46 | 0.885 | 0.861 | 0.766 | 80.00 | 0.800 | 0.714 | 0.613 |
| MACCS | 99.37 | 0.1057 | 0.994 | 0.99 | 0.987 | 84.69 | 0.847 | 0.884 | 0.691 | 89.47 | 0.895 | 0.733 | 0.790 |
| PubChem | 99.79 | 0.1449 | 0.998 | 0.995 | 0.996 | 80.08 | 0.801 | 0.810 | 0.599 | 75.78 | 0.758 | 0.717 | 0.515 |
| Substructure | 97.68 | 0.1702 | 0.977 | 0.97 | 0.953 | 78.57 | 0.786 | 0.745 | 0.560 | 78.94 | 0.789 | 0.711 | 0.580 |
| Substructure count | 99.78 | 0.144 | 0.998 | 0.995 | 0.996 | 79.41 | 0.794 | 0.750 | 0.577 | 82.1 | 0.821 | 0.777 | 0.641 |
| Klekota-Roth | 99.58 | 0.1349 | 0.996 | 0.99 | 0.991 | 83.64 | 0.836 | 0.796 | 0.664 | 83.15 | 0.832 | 0.782 | 0.664 |
| Klekota-Roth count | 99.58 | 0.137 | 0.996 | 0.99 | 0.991 | 81.97 | 0.82 | 0.796 | 0.631 | 87.36 | 0.874 | 0.869 | 0.747 |
| 2D atom pairs | 96.64 | 0.1955 | 0.966 | 0.965 | 0.931 | 85.74 | 0.860 | 0.854 | 0.713 | 84.21 | 0.842 | 0.822 | 0.683 |
| 2D atom pairs count | 99.79 | 0.1471 | 0.998 | 1 | 0.996 | 80.29 | 0.803 | 0.791 | 0.602 | 81.05 | 0.811 | 0.804 | 0.621 |

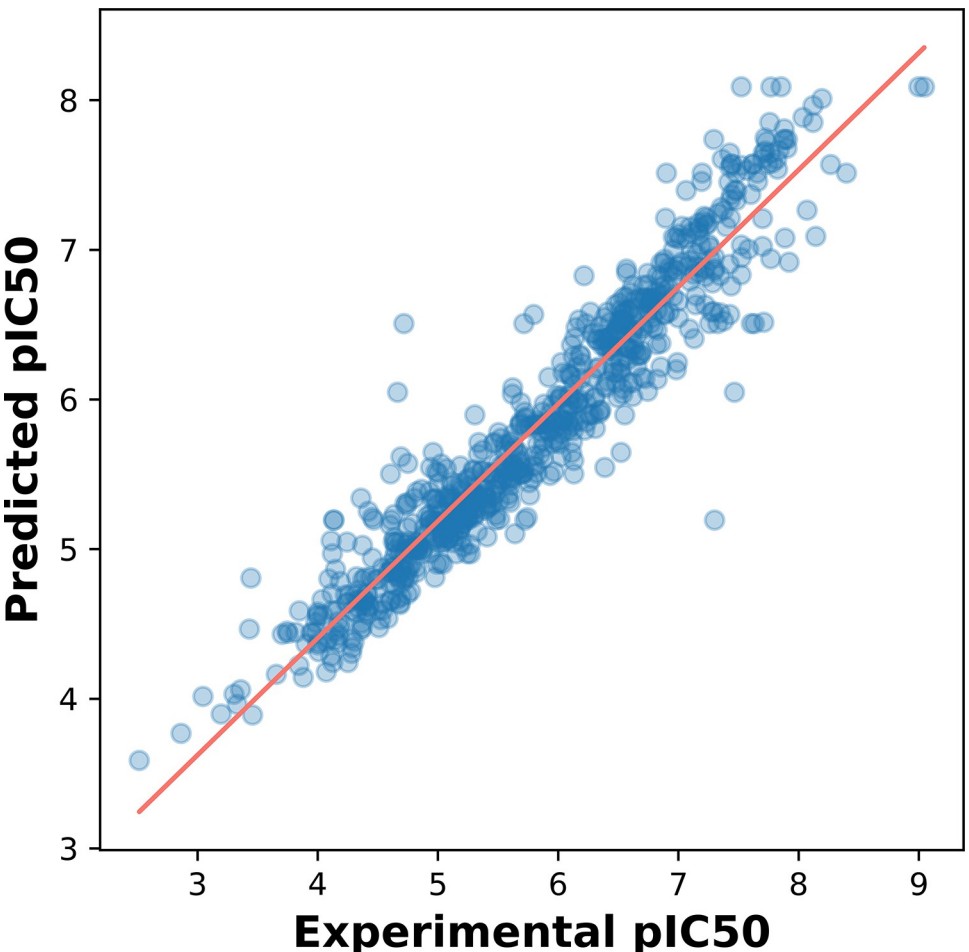

**Fig 3. Plot showing experimental versus predicted pIC50 values for model constructed with MACCS fingerprint descriptors.**

**Table 4. List of the top-ranking MACCS fingerprints and their corresponding description.**

| Fingerprint | SMARTS pattern | Substructure description |
|---|---|---|
| MACCSFP16 | (’[!#6;!#1]1~*~*~1’,0) | Heteroatom + any 2 heteroatoms + linkage to first atom |
| MACCSFP23 | (’[#7]~[#6](~[#8])~[#8]’,0) | NC(O)O |
| MACCSFP82 | (’*~[CH$_2$]~[!#6;!#1;!H0]’,0) | Any heavy atom + CH$_2$-heteroatom + H |
| MACCSFP87 | (’[F,Cl,Br,I]!@*@*’,0) | Halogen (part of chain) + any heteroatom connected to another heteroatom with an aromatic bond |
| MACCSFP89 | (’[#8]~*~*~[#8]’,0) | O + any 2 heteroatoms + O |
| MACCSFP103 | (’Cl’,0) | Cl |
| MACCSFP107 | (’[F,Cl,Br,I]~*(~*)~*’,0) | Halogen + heteroatom (heteroatom) heteroatom |
| MACCSFP115 | (’[CH$_3$]~*~[CH$_2$]~*’,0) | CH$_3$ + any heteroatom + CH$_2$ + any heteroatom |
| MACCSFP125 | (’?’,0) | Aromatic Ring > 1 |
| MACCSFP134 | (’[F,Cl,Br,I]’,0) | Halogen |
| MACCSFP145 | (’*1~*~*~*~*~*~1’,1) | 6-member ring > 1 |

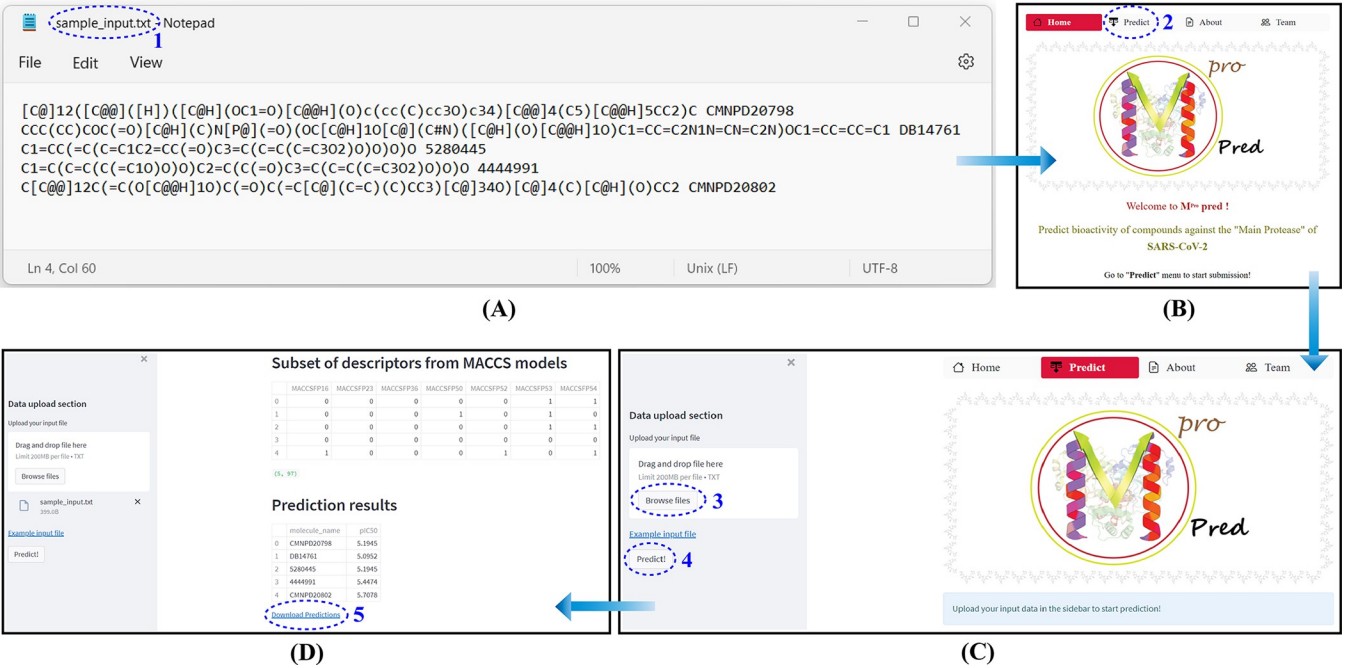

**Fig 4.**

## Model deployment as the M^pro^pred web-app and assessment

In order to allow biologists or chemists without a computer science background to apply the prediction model in their research, we deployed it as a public web-app known as the M^pro^pred and is available at https://share.streamlit.io/nadimfrds/mpropred/Mpropred_app.py. Briefly, a guide on using the M^pro^pred web-app (Fig 4) is given below:

**Step 1**. A text file (.txt) should be created containing the SMILES ID of the desired compounds space separated by a given name/ID (Fig 4A). SMILES IDs for any desired small compounds can be acquired from various databases e.g. Drugbank [54], PubChem [55] or ChemSpider [56] whereas custom compounds can be drawn on JSME structure editor [57] or ChemDraw [58] so as to create the SMILES notation of unknown compounds.

**Step 2**. The above-mentioned URL should be typed into any web browser to view homepage of the web-app (Fig 4B).

**Step 3**. The created text file should be uploaded to the web-app by clicking on the "Browse files" button (Fig 4C).

**Step 4**. The process of prediction can be started upon clicking on the "Predict!" button (Fig 4C).

**Step 5**. The results are showed in a box found below the "Prediction results" heading (Fig 4D). Typically, only a few seconds is required for the web-app to process the task. The users can also download the predicted results in the form of a CSV file by clicking the "Download Predictions" button.

## Binding affinity of CMNPD compounds with M^pro^

Out of the various possible binding positions of each compound predicted by Autodock Vina, the best one was picked considering the lowest binding energy. The molecular docking score of top 200 CMNPD compounds with M^pro^ ranged from -4.3 Kcal/mol to -10 Kcal/mol shown in S14 File while the result of top 5 compounds is presented in Table 5. The amino acid

**Table 5. Predicted pIC50 and binding affinity score of top 5 compounds from CMNPD database against M$^{pro}$.**

| Compound | Predicted pIC50 | Docking score (Kcal/mol) |
|---|---|---|
| CMNPD285 | 6.46 | -10.1 |
| CMNPD20581 | 7.00 | -9.6 |
| CMNPD12721 | 6.49 | -9.4 |
| CMNPD16005 | 6.37 | -9.4 |
| CMNPD6083 | 6.43 | -9.4 |

interactions of M$^{pro}$ with the top 5 compounds was also identified. The lowest binding energy was found for the compound CMNPD285. The CMNPD16005 is stabilized by a highest number of seven hydrogen bonds and four hydrophobic bonds while binding with the M$^{pro}$. The second highest number of hydrogen bonds (6) were formed in the CMNPD12721 complex which was also stabilized by seven hydrophobic bonds. All the 5 compounds formed stable interaction with the active site residues and the catalytic dyad comprised of His41 and Cys145 residues of M$^{pro}$. The detailed interaction profile of the top 5 compounds including the N3 ligand with M$^{pro}$ is explored in Fig 5.

## Molecular dynamics (MD) simulation results

The RMSD of backbone atoms of the protein-ligand complexes were analyzed to view their stability. It can be observed from Fig 6(A) that CMNPD16005 complex shows the lowest

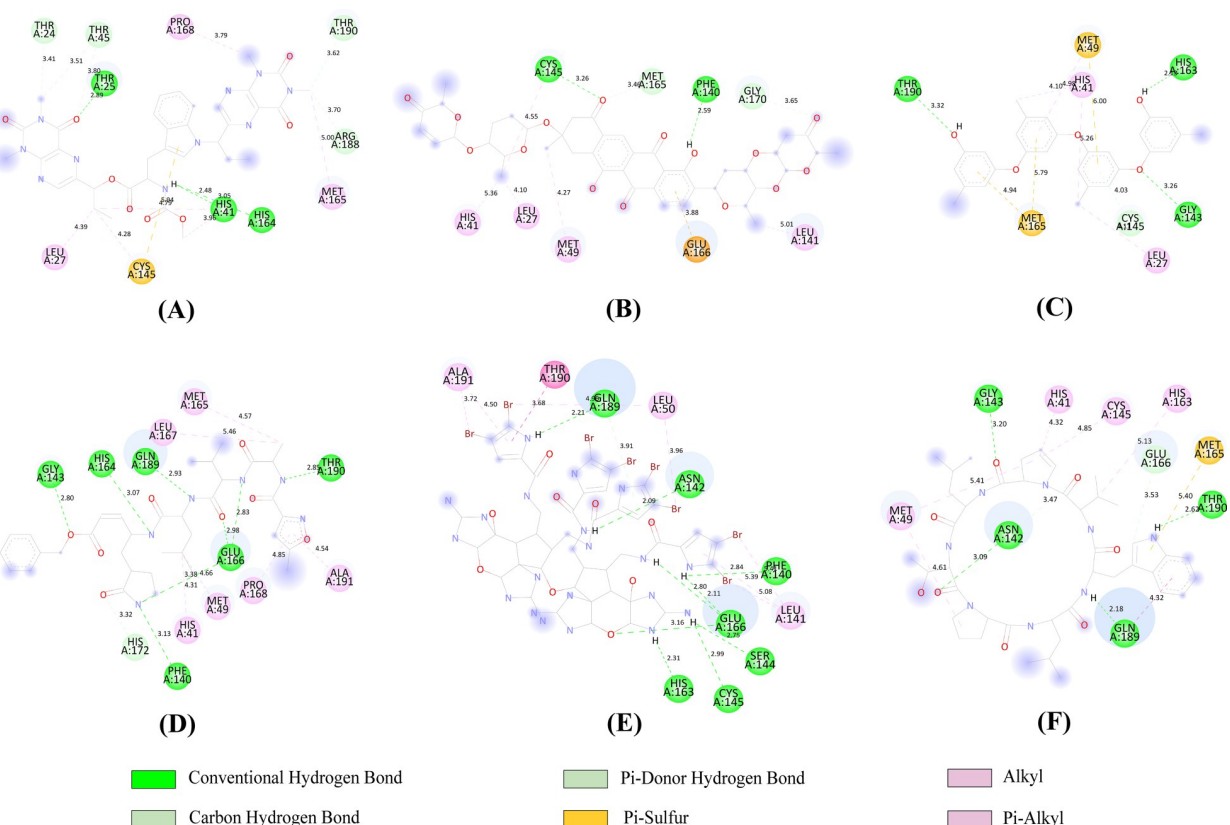

**Fig 5.** Two-dimensional (2D) representation of molecular docking analysis between the SARS-CoV-2 M$^{pro}$ and (A) N3, (B) CMNPD285, (C) CMNPD20581, (D) CMNPD12721, (E) CMNPD16005, (F) CMNPD6083.

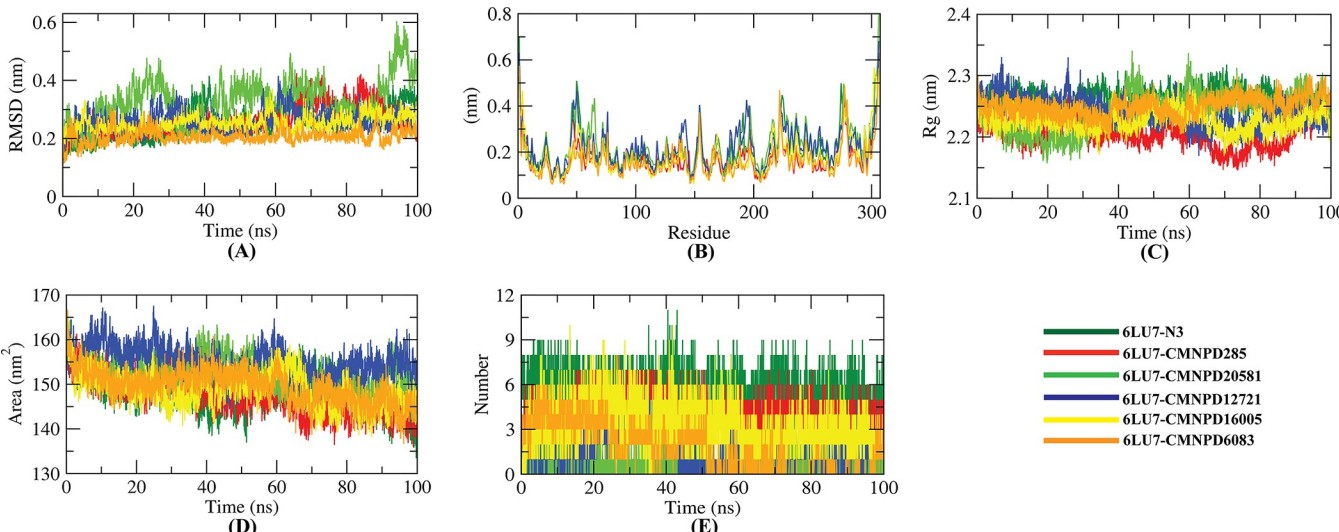

**Fig 6.** The Root-mean-square deviation (A), Root-mean-square fluctuation (B), Radius of gyration (C), Solvent accessible surface area (D) and hydrogen bond (E) analysis of protein-ligand complexes from the molecular simulation of 100 ns at 300 K.

RMSD than all other complexes. Surprisingly, the RMSD of the 6LU7-N3 complex is a bit higher than the CMNPD16005, which denotes the significant stability of CMNPD6083. The RMSD of CMNPD285 complex reaches to ∼0.4 nm from 60 to 85 ns, but the value increases after 85 ns and reaches to 0.3 nm. While viewing into the RMSD of CMNPD12721 complex, a steady increase of RMSD is observed after 60 ns. The value is decreased eventually indicating that CMNPD12721 might change the conformation of protein. Unlike the control and CMNPD16005 complex, RMSD of the CMNPD6083 complex is the mostly stable. Particularly, the CMNPD20581 complex shows the highest RMSD and higher degree of fluctuations throughout the period.

As RMSF aids in understanding the region of the receptor that is fluctuated throughout simulation, the flexibility of every residue is determined to gain a better understanding of how ligand binding impacts receptor flexibility. It is understood from Fig 6(B) that the binding of CMNPD12721 makes the M^pro most flexible in almost all areas in comparison to all other complexes. Overall, the residues such as Glu47, Met49, Leu50, Tyr154, Ala194, Thr196, Arg222, Asn277 and Phe305 are found flexible in case of both control and the ligand-bonded complexes.

The Rg represents the compactness of protein-ligand complexes. The lesser the fluctuation across the simulation period, the more compact the system is. The Rg of the 6LU7-N3 and CMNPD285 complexes were found to be nearly stable in case of fluctuation consistency throughout the simulation (Fig 6C). Besides, the Rg of CMNPD20581 was increased from 40 to 100 ns. The higher change of Rg might be due to protein folding, or distinct structural changes. The remaining complexes showed decreased Rg values indicating greater rigidity throughout the simulation period.

A higher SASA value implies that the protein volume is expanding, and a lower degree of fluctuation is mostly expected over time. SASA can be altered by the binding of any molecule, and this can have a significant impact on the receptor structure. The SASA values of all the complexes including the control were found lowest during the simulation period suggesting that the presence of these molecules potentially could limit protein expansion (Fig 6D).

Since intermolecular hydrogen bonds are known to contribute to conformational stability, the number of total hydrogen bonds in the protein-ligand complexes were determined. Most

**Table 6. Binding free energy calculations (MM/PBSA) for six protein-ligand complexes.**

| Complex | Van der Waal energy (kJ mol⁻¹) | Electrostatic energy (kJ mol⁻¹) | Polar solvation energy (kJ mol⁻¹) | SASA energy (kJ mol⁻¹) | Binding energy (kJ mol⁻¹) |
|---|---|---|---|---|---|
| 6LU7-N3 | -224.851 +/- 18.949 | -162.972 +/- 20.477 | 319.391 +/- 26.396 | -25.515 +/ 1.955 | -93.947 +/- 17.448 |
| CMNPD285 | -213.039 +/- 18.202 | -169.860 +/- 27.622 | 342.256 +/- 36.546 | -22.822 +/- 1.572 | -63.465 +/- 22.132 |
| CMNPD20581 | -230.604 +/- 15.446 | -7.170 +/- 3.876 | 91.983 +/- 11.163 | -19.485 +/- 1.290 | -165.277 +/- 14.898 |
| CMNPD12721 | -165.312 +/- 30.586 | -120.472 +/- 37.904 | 278.799 +/- 47.652 | -16.485 +/- 2.606 | -23.469 +/- 26.819 |
| CMNPD16005 | -174.597 +/- 15.161 | -599.461 +/- 34.553 | 499.675 +/- 37.798 | -21.811 +/- 1.399 | -296.193 +/- 25.797 |
| CMNPD6083 | -90.357 +/- 44.064 | -28.842 +/- 20.343 | 91.190 +/- 55.611 | -9.675 +/- 4.522 | -37.684 +/- 59.575 |

hydrogen bonds is observed for 6LU7-N3 complex, while the lowest number is observed in CMNPD20581 complex over the simulation period (Fig 6E). The remaining complexes possessed a significant number of hydrogen bonds (ranging from 3 to 8) compared to the CMNPD20581 complex.

## Post simulation binding free energy results

Using the MM/PBSA method, the binding free energies of the last 20 ns with a 100 ps interval was estimated from MD trajectories. The overall binding energies of all the complexes were negative, showing greater binding (Table 6). The CMNPD16005 complex showed the lowest binding free energy (-296.193 +/- 25.797 KJ/mol), indicating the most stable conformation of the compound. The other complexes similarly had a low binding energy, suggesting that they could be utilized as potential compounds. A comparative analysis of the binding free energies of the complexes were illustrated in Fig 7(A). The results for the amino acid residue contribution in the binding of the compounds are shown in Fig 7(B). The binding of the compounds to M^pro involved the notable contribution of leu27, Met49, Cys145, Leu167, Pro168, and Thr190 amino acid residues.

## Discussion

The COVID-19 pandemic has caused severe damage on the health and daily lives of billions of people around the world over the last two years. We've seen a race against time to vaccinate as many people as possible in recent months; however, discrepancies in vaccine distribution between nations, as well as new developing variants, pose an additional public health risk, making it difficult to achieve full immunization [59]. Several vaccine formulations are now available, assisting in the development of immunity [60–63]. Nonetheless, there is an increasing interest in developing new anti-covid medications. The M^pro, which is responsible for the cleavage of polypeptides during viral genome transcription, is a fascinating drug target in this scenario.

In the current study, we aimed to develop a classification model that is able to determine active from inactive compounds, and build a web-app for differentiating compounds for M^pro with selectivity. We followed the Organisation for Economic Co-operation and Development (OECD) recommendations to develop robust QSAR models for this purpose [64]. These guidelines comprise of the following major points: (i) the data set should have a defined endpoint, (ii) it should use an explicit learning algorithm, (iii) there should be a defined applicability domain of the built model, (iv) appropriate measurement of robustness and predictivity and (v) interpretation of the important features of the QSAR model. We initially extracted a dataset of 758 compounds from literature review and thresholds of <0.5 and >10 μM for identifying active compounds from the inactives in order to build a classification model. Upon

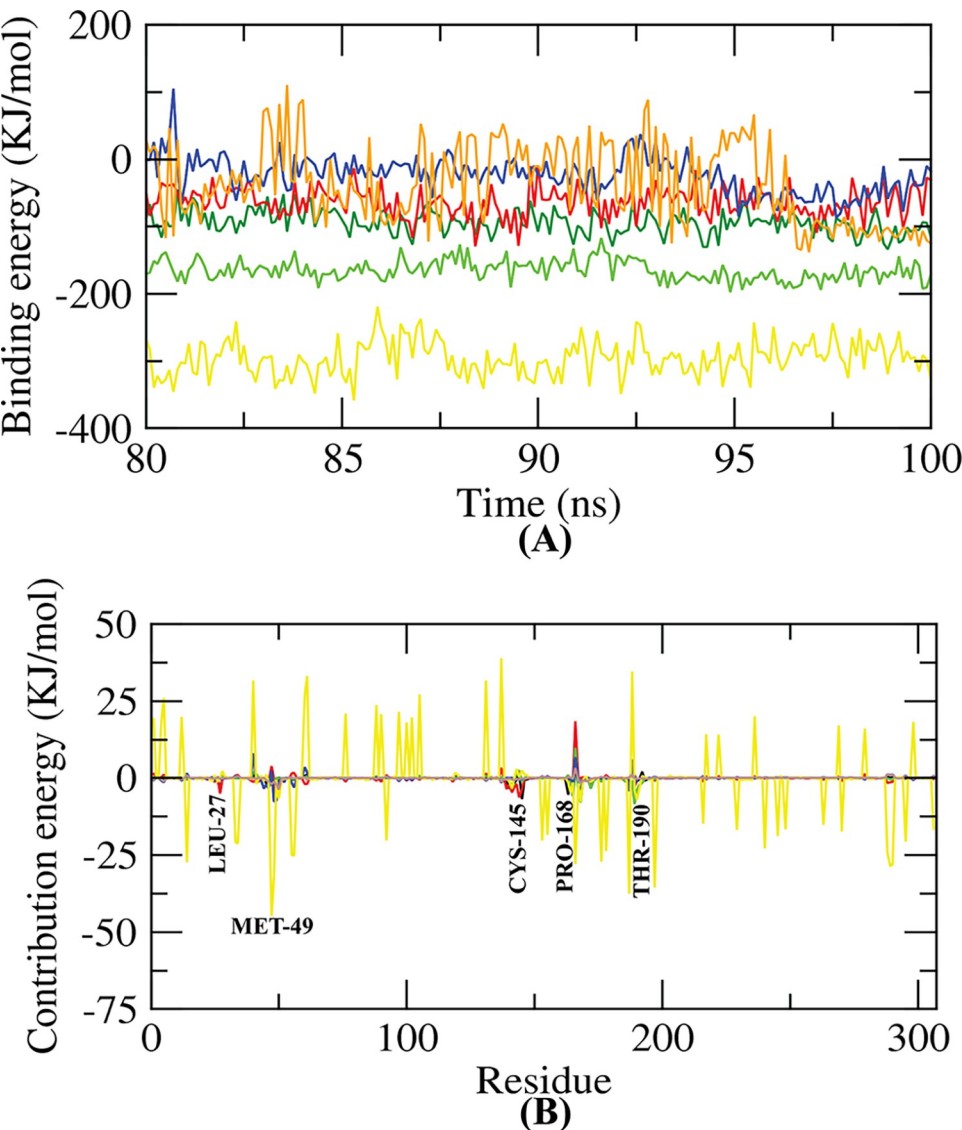

**Fig 7.** Graphical illustration of the binding free energy (A) and per residue contribution plot of protein-ligand complexes (B).

excluding the intermediate sets of compounds, we obtained a curated set of 478 compounds for detailed analysis. It is feasible to determine if a compound will exhibit the biological or pharmacological property needed for an orally active medicine in humans utilizing the Lipinski's rule-of-five (Ro5) approach. These characteristics are based on the fact that almost all drugs are relatively large lipophilic compounds with MW, ALogP, the number of hydrogen hydrogen bond donors, and the number of hydrogen bond acceptors. We found that most of the compounds meet the Ro5 criteria (Fig 2B–2E) and the findings of statistical analysis from Mann–Whitney U test showed a significant difference between the active and inactive compounds (Table 1). Also, the chemical space distribution shows that the external set lies well within the areas of the internal set indicating that the AD is well defined for the developed CSAR model found through PCA analysis results (Fig 2F).

Furthermore, we used interpretable molecular fingerprints to develop interpretable QSAR models and evaluated the model performances for all the used 12 fingerprints, following the aforementioned guidelines. Also, it is necessary to identify and address the activity cliffs in the data set using the data set's modelability score or MODI index before the predictive model can be developed. The data set was found to have a MODI value more than 0.65 for all the 12 fingerprint descriptors, indicating that it is reliable for developing a classification model (Table 2). Then we developed a QSAR model utilizing the random forest (RF) algorithm in order for differentiation of the active and inactive inhibitors for M$^{pro}$. The best averaged values determined by a 10-fold CV analysis was found for the MACCS fingerprint descriptors (Ac of 89%, Sn of 89%, Sc of 73%, and MCC of 79%) (Table 3). Similarly, Klekota–Roth and 2D atom pairs descriptors performed well, with the second and third highest best values for Ac and MCC, respectively, with Klekota–Roth fingerprints providing Ac and MCC values of 83.64% and 0.664, respectively, and 2D atom pairs fingerprints providing Ac and MCC values of 85.74% and 0.713, respectively (Table 3). We found that the MACCS fingerprints were the best choice for model interpretation based on the Ac values, MCC values, overall external and CV.

Later, an investigation of the important features on selected descriptors was conducted to obtain a better view of the mechanistic details driving M$^{pro}$. The top-ranked MACCS descriptors include descriptors of various classes such as aromatic compounds, nitrogen-containing compounds, oxygen-containing compounds and halogens as obtained from the RF model (Table 4). M$^{pro}$ has been shown to be inhibited by a range of N-substituted isatin derivatives, with the highest activity being associated with derivatives having carboxamide groups at C-5 of the isatin core (IC50 = 0.045–17.8 μM) [65]. Several oxygen atoms containing small compounds were also found to inhibit M$^{pro}$ and blocks viral transcription [66,67]. Kowit et al. identified halogenated baicalein as a potent inhibitor of the M$^{pro}$ and they confirmed its inhibitory activity in an in vitro assay [68]. It was also found that the addition of halogen groups improves binding strength by an order of magnitude [69]. Hossum et al. generated a pharmacophore model and found three acceptor features and one aromatic ring feature as common in all the active hits including the co-crystallized ligand [70]. Thus, the top-ranked MACCS descriptors are in significant correlation with the properties of laboratory validated potent M$^{pro}$ inhibitors.

In a normal predictive model life cycle, after models are validated and outcomes are shown in the publications, the model's utility is essentially over. In this way, the model has accomplished its goal to make predictions and offer useful insights into the underlying key characteristics. We believed that deployment of the predictive model as a public web-app that allows scientists and researchers, particularly in the fields of computational chemistry and biology, to use the predictive insights from the model would significantly improve its value, while also benefiting scientific communities, would greatly extend the model's life cycle. We made the web-app available at "Streamlit share" platform (Fig 4). In order to test the web-app to determine the correlation between predicted pIC50 and the binding affinity, we applied an integrated molecular modeling approach. All the available 31,492 compounds were submitted to the web-app to predict their pIC50 and it was found that top five compounds with highest binding affinity to M$^{pro}$ had pIC50 values ranging from 6.37 to 7 (Table 5). They formed sufficient hydrogen bond and hydrophobic interactions and all of them formed stable interactions with the catalytic dyad consisting of His41 and Cys145 (Fig 5).

Also, MD simulation results re-confirmed the stability of these five compounds with M$^{pro}$. The RMSD plot indicates that all the five compounds are stable, with no unexpected rises in RMSD values across the simulated time (Fig 6A). The complexes had fewer fluctuations in the allowed range, according to the RMSF study (Fig 6B). The radius of gyration (Rg) of the protein-ligand complexes tended to be similar, indicating that every complex had a similar

compactness behavior (Fig 6C). The SASA values showed that the volume of the complexes did not substantially increase (Fig 6D). Throughout the simulation, a significant number of hydrogen bonds were observed in all of the complexes, further elucidating their conformational stability (Fig 6E). Furthermore, the binding free energies for all of the complexes were estimated using the MM/PBSA method, and the results suggest that the complexes have a favorable binding energy with M$^{pro}$ (Table 6 and Fig 7A). It can be determined from the per-residue interaction energy profile that the leu27, Met49, Cys145, Leu167, Pro168, and Thr190 residues of M$^{pro}$ played an important role in protein-ligand stability and contributed significantly to the binding of the compounds (Fig 7B). As a result, these compounds may have the potential to interfere with and block the activity of SARS-CoV-2 M$^{pro}$.

Thus, the web-app presented in the current study can be utilized for further research on various compounds to get a view into their anti-M$^{pro}$ activity. Also, upon evaluating the toxicity of the five marine derived compounds by various toxicity assays, their inhibition efficacy can be tested through in vitro laboratory validations.

## Supporting information

**S1 File. The SMILES ID and additional details of the 758 compounds.**
(XLSX)

**S2 File. Computed AtomPairs2DCount fingerprint of our dataset.**
(CSV)

**S3 File. Computed AtomPairs2D fingerprint of our dataset.**
(CSV)

**S4 File. Computed CDK fingerprint of our dataset.**
(CSV)

**S5 File. Computed CDK Extended fingerprint of our dataset.**
(CSV)

**S6 File. Computed CDK Graph only fingerprint of our dataset.**
(CSV)

**S7 File. Computed EState fingerprint of our dataset.**
(CSV)

**S8 File. Computed KlekotaRothCount fingerprint of our dataset.**
(CSV)

**S9 File. Computed KlekotaRoth fingerprint of our dataset.**
(CSV)

**S10 File. Computed MACCS fingerprint of our dataset.**
(CSV)

**S11 File. Computed PubChem fingerprint of our dataset.**
(CSV)

**S12 File. Computed Substructure fingerprint of our dataset.**
(CSV)

**S13 File. Computed SubstructureCount fingerprint of our dataset.**
(CSV)

**S14 File. The docking score of all the CMNPD compounds with M$^{pro}$.**
(XLSX)

## Acknowledgments

The author(s) acknowledge the Bioinformatics Division, National Institute of Biotechnology, Bangladesh, for their extended support of supercomputing system during this study.

## Author Contributions

**Conceptualization:** Nadim Ferdous, Mahjerin Nasrin Reza, Suhami Napis, Kamal Chowdhury, A. K. M. Mohiuddin.

**Data curation:** Nadim Ferdous, Mahjerin Nasrin Reza.

**Formal analysis:** Nadim Ferdous, Mahjerin Nasrin Reza.

**Investigation:** Mahjerin Nasrin Reza, Shahin Mahmud.

**Methodology:** Nadim Ferdous, Mahjerin Nasrin Reza.

**Software:** Mahjerin Nasrin Reza.

**Supervision:** Mohammad Uzzal Hossain, Shahin Mahmud, Suhami Napis, Kamal Chowdhury, A. K. M. Mohiuddin.

**Validation:** Mohammad Uzzal Hossain, Shahin Mahmud, A. K. M. Mohiuddin.

**Writing – review & editing:** Mohammad Uzzal Hossain, Shahin Mahmud, Kamal Chowdhury, A. K. M. Mohiuddin.

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
