## [Decision Letter · Decision Letter 0]

12 Jan 2023

PONE-D-22-21799Mpropred: A machine learning (ML) driven Web-App for bioactivity prediction of SARS-CoV-2 main protease (Mpro) antagonistsPLOS ONE

Dear Dr. Mohiuddin,

Thank you for submitting your manuscript to PLOS ONE. After careful consideration, we feel that it has merit but does not fully meet PLOS ONE’s publication criteria as it currently stands. Therefore, we invite you to submit a revised version of the manuscript that addresses the points raised during the review process. Please address the comment of the reviewer, especially the question about the meaning of the fingerprints. Please consider including a figure with a 3D structure of the Mpro receptor site to better convey the structural discussion in the introduction and results. Isabela et al should be probably referenced as Gomes et al. on line 89. Many Mpro inhibitors, including N3 in the structure used for the MD simulations, are covalent inhibitors. It would be beneficial to discuss whether the inhibitors identified by the model are likely covalent inhibitors and which fingerprints recognize such inhibitors.     Please submit your revised manuscript by Feb 26 2023 11:59PM. If you will need more time than this to complete your revisions, please reply to this message or contact the journal office at plosone@plos.org. Please include the following items when submitting your revised manuscript:A rebuttal letter that responds to each point raised by the academic editor and reviewer(s). You should upload this letter as a separate file labeled 'Response to Reviewers'.A marked-up copy of your manuscript that highlights changes made to the original version. You should upload this as a separate file labeled 'Revised Manuscript with Track Changes'.An unmarked version of your revised paper without tracked changes. You should upload this as a separate file labeled 'Manuscript'.

We look forward to receiving your revised manuscript.

Kind regards,

Emilio Gallicchio, Ph.D.

Academic Editor

PLOS ONE

and https://journals.plos.org/plosone/s/file?id=ba62/PLOSOne_formatting_sample_title_authors_affiliations.pdf.

Reviewers' comments:

Reviewer's Responses to Questions

**Comments to the Author**

1. Is the manuscript technically sound, and do the data support the conclusions?

Reviewer #1: Yes

2. Has the statistical analysis been performed appropriately and rigorously? 

Reviewer #1: Yes

3. Have the authors made all data underlying the findings in their manuscript fully available?

Reviewer #1: Yes

4. Is the manuscript presented in an intelligible fashion and written in standard English?

Reviewer #1: Yes

5. Review Comments to the Author

Reviewer #1: Peer Review of:

Mpro 1 pred: A machine learning (ML) driven Web-App for bioactivity prediction of SARS-CoV-2 main protease (Mpro) antagonists

The manuscript, written by Nadim Ferdous and Mahjerin Nasrin Reza et. al discusses about the development of a prediction model to classify inhibitors and non-inhibitors of SARS CoV-2 Main Protease(Mpro) which is a potential drug target to control Coronavirus pandemic that emerged in 2019. The study has its merits and have a great potential to help researchers in the field to identify effective compounds and design specific inhibitors. The authors also developed a web-app to run the model and make it accessible to the research community.

The authors validated the model by screening existing datasets, and conducting molecule docking, MD simulations and binding free energy estimates. However, the concept underlying the prediction of binding free energies on the select hits , seemed to have some limitation which requires further explanation.

In line 241 : the authors mentioned “g_mmpbsa” package to compute binding free energies. This was also mentioned in the “Results” section. However, the article cited for “g_mmpbsa” states about computing binding energies of the complex and does not include the entropic terms as stated in the Github page of the tool (https://github.com/RashmiKumari/g_mmpbsa).

The authors also mentioned “binding energy” and “binding free energy” simultaneously to refer to the same quantity that was measured by the “g_mmpbsa” tool.

Between Line 384 and 387, the authors mentioned that CMNPD16005 is predicted to have the lowest binding free energy of around -296.193 KJ/mol, but in Table 6, it was recorded under the column “Binding energy”. It would be appropriate if the authors use a single term for the quantity measured in this study.

Table 3 (Docking score) and Table 6 (Binding free energies) are reported with different units (Kcal/mol and KJ/mol respectively). For the ease of reading the article, the authors should use the same unit and conversion, if required, for all data that is tabulated in the study.

It would be great if the authors explain a little bit more about the difference between binding energy and binding free energies and the type of data that was provided in Table 6. Based on what authors explained about the use of “g_mmpbsa” tool, the data in the last column of Table 6 , is likely to be the interaction energy and not binding free energy that was mentioned in the text.

For conducting more reliable binding free energy calculations, it will be great if the authors choose a robust method like FEP (Free-Energy Perturbation), or other alchemical approaches and compare the results with the experimental affinities, as the model holds a lot of potential, so, a more robust physics-based prediction approach will be a great addition to this study.

Line 133 : "…CDK (including extended and graph only version), E-state, PubChem, Klekota–Roth, Substructure and MACCS ": the authors should describe the acronyms in this section, though it seems like these are different fingerprints that are used to describe the structures from different classes. A reference to Table 2 where authors have explained different fingerprints, will be appropriate for the reader to get an idea.

Few other minor modification and edits may help the readers to understand the manuscript more clearly.

Line 117 , subheading in Materials and Methods, repeated again in Line 118

Line 124 : “…with IC50 values that ranged between 1 and 10 were excluded ..” : the authors should mention the concentration units

Line 135 to 142 : the author mentioned “variables” and “constants” when mentioning about fingerprint descriptors. The authors should explain more about these features as the molecular fingerprinting seems to be a crucial step in the overall design of the model.

Line 264: The author mentioned “AD” here but did not explain what the acronym means.

Line 333 : Did the author meant to say “hydrophobic interactions” instead of "hydrophobic bonds" throughout the manuscript?

In Table 5, units are missing for the IC50 values.

6. PLOS authors have the option to publish the peer review history of their article (what does this mean?). If published, this will include your full peer review and any attached files.

Reviewer #1: No

---

## [Author Response · Author response to Decision Letter 0]

26 May 2023

May 23, 2023

To 

Editor-in-Chief

PLOS One

Re: Resubmission of research article (Submission ID: PONE-D-22-21799)

Dear Sir,

Thank you very much for sending us your suggestions and comments. 

We went through the invaluable comments of the honourable editor and tried our best to answer, explain and change the manuscript accordingly. We have addressed the queries each by each in details and listed in the following pages. With the changes made we belief the manuscript is improved substantially. 

We hope that the revised manuscript will be evaluated fairly and considered for publication in your esteemed journal. 

Sincerely yours, 

Dr A. K. M. Mohiuddin

Professor

Department of Biotechnology and Genetic Engineering

Mawlana Bhashani Science and Technology University, Santosh, Tangail-1902, Bangladesh

E-mail: akmmohiu@yahoo.com

Authors Response to honourable Editor Comments

Editor comments

In line 241: the authors mentioned “g_mmpbsa” package to compute binding free energies. This was also mentioned in the “Results” section. However, the article cited for “g_mmpbsa” states about computing binding energies of the complex and does not include the entropic terms as stated in the Github page of the tool (https://github.com/RashmiKumari/g_mmpbsa).

The authors also mentioned “binding energy” and “binding free energy” simultaneously to refer to the same quantity that was measured by the “g_mmpbsa” tool.

Author Response:

Thank you very much for your suggestions that we feel our manuscript got incredibly improved. We have changed the “binding energy” term to “docking energy” in our revised manuscript. 

Between Line 384 and 387, the authors mentioned that CMNPD16005 is predicted to have the lowest binding free energy of around -296.193 KJ/mol, but in Table 6, it was recorded under the column “Binding energy”. It would be appropriate if the authors use a single term for the quantity measured in this study.

Author Response:

Thank you again for your valuable comment. We have changed the title of the last column in Table 6 to “Binding free energy” in our revised manuscript. 

Table 3 (Docking score) and Table 6 (Binding free energies) are reported with different units (Kcal/mol and KJ/mol respectively). For the ease of reading the article, the authors should use the same unit and conversion, if required, for all data that is tabulated in the study.

Author Response:

Thank you again for your valuable comment. Binding free energy is the equilibrium constant of complex formation, laws of thermodynamics relate it to the change in Gibbs free energy upon binding: The change in Gibbs free Energy of Binding is equal to - RT times the natural logarithm of the Equilibrium constant, where R is the gas constant and T the absolute temperature. These are experimentally determinable parameters that we predict by calculation. The Docking score is a computational result that is specific for a particular program and energy function, and that in an ideal case allows to predict binding free energy and binding affinity, or to at least rank different complexes according to those parameters. This is why we kept different units for the two different scores.

It would be great if the authors explain a little bit more about the difference between binding energy and binding free energies and the type of data that was provided in Table 6. Based on what authors explained about the use of “g_mmpbsa” tool, the data in the last column of Table 6, is likely to be the interaction energy and not binding free energy that was mentioned in the text.

Author Response:

Thank you again for your valuable comment. The method that the “g_mmpbsa” tool uses, the MM/PBSA method, is widely used in the calculation of receptor-ligand binding free energy. The full name of this method is Molecular Mechanics/Poisson Boltzmann Surface Area. As the name suggests, this method splits the binding free energy into molecular mechanics terms and solvation energies to be calculated separately. The basic principle is to calculate the difference between the binding free energy of two solvated molecules in the bound and unbound states or to compare the free energy of different solvation conformations of the same molecule. We have changed the “binding energy” term to “docking energy” and kept the “binding free energy” term only in case of the g_mmpbsa tool in our revised manuscript.

For conducting more reliable binding free energy calculations, it will be great if the authors choose a robust method like FEP (Free-Energy Perturbation), or other alchemical approaches and compare the results with the experimental affinities, as the model holds a lot of potential, so, a more robust physics-based prediction approach will be a great addition to this study.

Author Response:

Thank you again for your valuable comment. We agree with the authors that addition of the Free-Energy Perturbation method to calculate binding free energy calculations would be a great addition to our study. Unfortunately, due to lack of supercomputing power, we are unable to perform that at this moment.

Line 133: "…CDK (including extended and graph only version), E-state, PubChem, Klekota–Roth, Substructure and MACCS ": the authors should describe the acronyms in this section, though it seems like these are different fingerprints that are used to describe the structures from different classes. A reference to Table 2 where authors have explained different fingerprints, will be appropriate for the reader to get an idea.

Author Response:

Thank you again for your valuable comment. We have added the available acronyms of the fingerprints in Table 2 of our revised manuscript.

Few other minor modification and edits may help the readers to understand the manuscript more clearly.

Line 117, subheading in Materials and Methods, repeated again in Line 118

Author Response:

Thank you again for your valuable comment. We removed the repeated line in Line 118 in the revised manuscript.

Line 124: “…with IC50 values that ranged between 1 and 10 were excluded ..” : the authors should mention the concentration units

Author Response:

Thank you again for your valuable comment. We added the concentration unit (μM) of the IC50 values in the revised manuscript.

Line 135 to 142: the author mentioned “variables” and “constants” when mentioning about fingerprint descriptors. The authors should explain more about these features as the molecular fingerprinting seems to be a crucial step in the overall design of the model.

Author Response:

Thank you again for your valuable comment. We added the citations of the publications that used the method we followed in selecting the fingerprint descriptors in the revised manuscript

Line 264: The author mentioned “AD” here but did not explain what the acronym means.

Author Response:

Thank you again for your valuable comment. We mentioned the acronym of AD in line 195-196.

Line 333: Did the author meant to say “hydrophobic interactions” instead of "hydrophobic bonds" throughout the manuscript?

Author Response:

Thank you again for your valuable comment. We meant "hydrophobic interactions" instead of "hydrophobic bonds" throughout the manuscript.

In Table 5, units are missing for the IC50 values.

Author Response:

Thank you again for your valuable comment. We added the units for the IC50 values in Table 5 of our revised manuscript.

---

## [Editor Report · Decision Letter 1]

1 Jun 2023

Mpropred: A machine learning (ML) driven Web-App for bioactivity prediction of SARS-CoV-2 main protease (Mpro) antagonists

PONE-D-22-21799R1

Dear Dr. Mohiuddin,

We’re pleased to inform you that your manuscript has been judged scientifically suitable for publication and will be formally accepted for publication once it meets all outstanding technical requirements.

Kind regards,

Emilio Gallicchio, Ph.D.

Academic Editor

PLOS ONE
---

## [Editor Report · Acceptance letter]

8 Jun 2023

PONE-D-22-21799R1 

M^pro^pred: A machine learning (ML) driven Web-App for bioactivity prediction of SARS-CoV-2 main protease (M^pro^) antagonists 

Dear Dr. Mohiuddin:

I'm pleased to inform you that your manuscript has been deemed suitable for publication in PLOS ONE. Congratulations! Your manuscript is now with our production department. 

Kind regards, 

on behalf of

Dr Emilio Gallicchio 

Academic Editor

PLOS ONE